# Emergency Presentations of Pediatric Sickle Cell Disease in French Guiana

**DOI:** 10.3390/diseases13050142

**Published:** 2025-05-04

**Authors:** Carine Fankep Djomo, Souam Nguele Sile, Narcisse Elenga

**Affiliations:** 1Department of Paediatrics and Surgery, Cayenne Hospital, 3 Avenue Alexis Blaise, Cayenne 97300, French Guiana; cadjomo@gmail.com; 2Department of Pediatrics, University of N’Djamena, Campus Gardolé, Avenue Mobutu, N’Djamena BP 1117, Chad; souamsile@yahoo.ca; 3Sickle Cell Reference Centre, Cayenne Hospital, 3 Avenue Alexis Blaise, Cayenne 97300, French Guiana; 4Faculty of Medicine Hyacinthe BASTARAUD, University of the French West Indies, Pointe-à-Pitre BP 145 97154, Guadeloupe; 5University of French Guinea, Faculty of Health, Campus De Trou Biran, 2091 route Baduel, Cayenne 97300, French Guiana

**Keywords:** sickle cell disease, children, emergency visits, vaso-occlusive crisis, medical education

## Abstract

Background/Objectives: This study aimed to estimate the proportion of pediatric emergency admissions related to sickle cell disease. Methods: This is a cross-sectional study. The data were collected over a period of 9 years, from 1 January 2014 to 31 December 2022. Results: We recorded 858 emergency department visits related to sickle cell disease out of a total of 135,000 pediatric emergency department visits, giving a prevalence of 6.4 per 1000 children aged up to 18 years. The median age was 12 years (8–16) years. The average waiting time in the emergency department for children with sickle cell disease was 2 h (±1) in 2014 and 45 min (±15) in 2022. Children with sickle cell anemia were more likely than others to have been seen by a consultant in an emergency department. The most commonly associated pathology was asthma, with a frequency of 17%. The risk factors for hospitalization were an age between 5 and 10 years and a severe form of sickle cell disease. Conclusions: The treatment of pain and fever were often delayed. This leads us to suggest that systematic prior communication between the pediatric hematologist and the emergency physician is crucial. However, there is a need to define best practices for the management of children with sickle cell disease presenting to the emergency department with a fever.

## 1. Introduction

Sickle cell disease (SCD), an autosomal recessive disorder, is the most common and serious genetic disorder worldwide [1]. Ranked as the 4th public health priority in France [2], SCD is caused by the production of abnormal hemoglobin due to a single mutation on chromosome 11. In regard to this mutation, glutamic acid is replaced by valine, resulting in the replacement of globin beta chains (also known as A chains) by S chains with polymerization capacity.

There are several types of SCD. Sickle cell anemia (SS or SCA): When a child inherits a substitution beta globin gene (the sickle cell gene) from each parent, the child has SCA. Populations that have a high frequency of SCA are those of African and Indian descent. Sickle hemoglobin C disease (SC): Individuals with SC have a slightly different substitution in their beta globin genes that produces both hemoglobin C and hemoglobin S. SC disease can cause symptoms similar to SCA, but patients have less anemia due to a higher blood count. Populations with a high incidence of SC disease are those of West African, Mediterranean, and Middle Eastern descent. Sickle beta thalassemia (Sß-thalassemia): Individuals with Sß-thalassemia also have substitutions in both of their beta globin genes. The severity of the disease varies according to the amount of normal beta globin produced. If no beta globin is produced (sickle beta zero thalassemia or Sß°thalassemia), the symptoms are almost identical to those of SCA, with severe cases requiring chronic blood transfusions. If beta globin is produced (sickle beta plus thalassemia or Sß+thalassemia), the symptoms are moderate. Populations with a high incidence of Sß thalassemia are those of Mediterranean and Caribbean descent. It is common to classify SS and Sß°thalassemia as SCA [1,2].

French Guiana (a French overseas territory located in South America) is dominated by the Afro-Caribbean population. Sandwiched between Brazil on one side and the little-known Suriname on the other, French Guiana is the only European territory in the Amazon; with about 50% of its population living below the poverty line, this territory is home to most endemic and/or epidemic infections, such as malaria, leishmaniasis, Chagas disease, histoplasmosis, and dengue fever [3,4]. Among French Guiana’s neighbors, only Brazil has a structured programme for neonatal screening and the management of sickle cell disease. In French Guiana, 10% of the population carried the sickle cell gene. Each year, more than 30 children are born with SCD in the region, or approximately one in every 227 births [3,4]. Those individuals with SS homozygotes, approximately 60% of patients in French Guiana, develop the most severe form of the disease. Other forms of SCD, in which hemoglobin S is combined with another abnormal hemoglobin, are also severe. These mutations are known as double heterozygotes and are mainly the SC form (approximately 37% in French Guiana) and the Sß-thalassemia form (3%). These different forms do not have the same clinical severity, basal hemoglobin concentration, or life expectancy [5].

Sickle cell hemoglobin (HbS) molecules, in their deoxygenated form, polymerize to form intracellular crystals that deform the red blood cell, giving it a characteristic sickle (or banana) shape, the sickle cell. The deformed red blood cells lose their elasticity and need to pass through the microcirculation. Deformed red blood cells are destroyed more quickly than normal red blood cells, which explains the occurrence of hemolytic anemia. The stiffening and deformation of red blood cells and the increase in blood viscosity explain the vaso-occlusive complications of the disease, especially as sickle red blood cells adhere to the vascular endothelium. SCD combines three main categories of clinical manifestations, with the extent of the variability in the clinical expression dependent on the individual: chronic hemolytic anemia with episodes of acute deterioration, vaso-occlusive phenomena, and susceptibility to bacterial infections [6].

SCD severity is related to the occurrence of complications, which are the cause of repeated consultations. Early recognition and appropriate management of these complications determines the prognosis of the disease.

The aim of our study was to estimate the proportion of pediatric emergency admissions related to SCD and to describe the risk factors for hospitalization following emergency admission.

## 2. Patients and Methods

### 2.1. Study Setting

French Guiana is an overseas French department and region (DROM) in South America, with the city of Cayenne as its capital. French Guiana is located off the northeastern coast of South America. It is bordered to the south and east by the Brazilian state of Amapa, and to the northeast by the Atlantic Ocean. The Maroni River separates French Guiana from Suriname to the west. The population is very diverse and comprises Creoles, a minority of French metropolitan areas, Brazilians, Surinamese, Haitians, other Caribbean peoples, Chinese, Laotians, and the Bushinengue (Maroon) community. The French National Institute for Statistics and Economic Studies (INSEE) estimated that the population of French Guiana would reach 301,999 on 1 January 2023 [7]. Due to its geographical location, the weather in French Guiana is equatorial. However, the year is divided into four seasons:The long dry season runs from mid-July until the end of November;December to February is a short rainy season;March to mid-April is a short summer;The main rainy season occurs from mid-April until the end of June.

Because of this equatorial climate, the temperature in French Guiana varies little throughout the year, neither in regard to the air or water, or between day and night. It generally remains between 25 °C and 33 °C. During the long dry season, rain is rare, and temperatures often exceed 30 °C, while humidity falls below 50%. The wettest months are May and June. An average rainfall of 500 mm is expected (Météo France). Le climat guyanais, http://pluiesextremes.meteo.fr/guyane/Le-climat-guyanais.html accessed on 2 January 2025.

The Center Hospitalier de Cayenne (CHC) plays a central role in this region. It has 795 beds and places and accounts for more than 60% of all hospital stays in the region.

The reference center for the management of SCD and the pediatric emergency department (ED) at the CHC served as the study setting.

Opened in September 2014, the sickle cell reference center for children and adults with SCD is open every day from 7 am to 5 pm, Monday to Friday. It welcomes patients by appointment and sickle cell emergencies. Outside the opening hours, emergency patients are admitted to the emergency department. The center also has a day hospital and an accredited sickle cell education program. This center is also responsible for the screening, surveillance, counselling, and prenatal diagnosis of SCD. Its multidisciplinary staff include physicians, nurses, social workers, psychologists, and counsellors. It also offers regular follow-ups, annual check-ups, and universal screening for cerebral vascular disease for children with SCA.

### 2.2. Type of Study

This is a cross-sectional study. The data were collected over a period of 9 years, from 1 January 2014 to 31 December 2022.

### 2.3. Study Population

All the sickle cell patients were aged between 0 and 18 years and presented to the ED during the study period.

Inclusion criteria:

Sickle cell patients aged less than or equal to 18 years at the time of consultation, residing in French Guiana, with access to health insurance, whose medical file was complete, and who presented at the sickle cell disease reference center or the pediatric emergency department for an acute complication during the study period, were included in the study.

Non-inclusion criteria; the following patients were not included in our study:

Patients without sickle cells;

-Patients living outside French Guiana;

Patients who refused to participate in the study.

The study variables are as follows: Sociodemographic variables:-Age;-Sex.

Clinical variables:-Medical history: type of SCD, history of >3 annual vaso-occlusive crises (VOCs), acute chest syndrome (ACS), stroke, splenectomy, cholecystectomy;-Associated chronic pathology: asthma, autoimmune disease, obstructive sleep apnea syndrome (OSAS), Gougerot–Sjögren syndrome, osteonecrosis of the femoral head, hypertension;-Date of consultation;

Reasons for consultation;

-Length of hospital stay: treatment data;

Treatment: hydroxyurea, transfusion exchange (ET), blood transfusion, oxygen, antibiotics, hydration and analgesics, morphine.

### 2.4. Data Collection

The data required for our study were collected from paper and computerized medical records (Cora and HM software 2017), ED consultation records, ED and hospital admission reports, and were entered into an EXCEL file.

### 2.5. Statistical Analysis

The data were recorded in an EXCEL file. The statistical analyses were performed using STATA 16.0 software (Stata Corp LP, College Station, TX, USA). Quantitative variables were described as the mean and standard deviation or median and interquartile range (IQR), depending on the distribution of the variable, as well as the minimum and maximum values. Qualitative variables were expressed as numbers and percentages. In the statistical analysis, we compared the sociodemographic, clinical, and outcome characteristics of patients who were admitted to hospital after being admitted to the emergency department with those who were discharged, i.e., not admitted to hospital. This allowed us to perform a bivariate analysis using logistic regression. The variables that were statistically significant in the bivariate analysis were used in the multivariate analysis. 

### 2.6. Ethical and Regulatory Status

The typology of this study corresponds to “Not Human Subjects Research”. All the data were collected from the medical records of patients treated by the pediatric ED and the sickle cell reference center. The data were pseudonymized and processed by the medical staff in these departments. The study is therefore internal research, as defined by the “Commission Nationale de l’ Informatique et des libertés (CNIL)”. In addition, the participants were informed collectively through posters in the pediatric ED and sickle cell center, in the welcome booklet, and on the hospital website (general information on clinical research). Patients’ objections to participating in the study were considered. We also obtained parents’ consent to publish their children’s clinical data. The study was registered in the hospital’s data processing register by the hospital data protection officer at the Center Hospitalier de Cayenne.

## 3. Results

From 1 January 2014 to 31 December 2022, we recorded 858 ED visits by children with SCD out of a total of 135,000 pediatric emergency department visits, giving a prevalence of 6.4 per 1000 children aged up to 18 years. In total, 148 children were included in this study.

The median age was 12 years, with interquartile ranges (between 25% and 75%) of 8–16 years. The mean age was 11.5 years, with a standard deviation of ±5 years. The over 10 years age group was the most represented in the sample, with 62% of patients being within this age group (Table 1). Male and female patients were almost equally represented, with 441/416 (Table 1). The average waiting time in the ED for children with SCD was 2 h (±1) in 2014 and 45 min (±15) in 2022. Children with SS and Sβ°thalassemia genotypes were more likely than others to have seen a consultant in the context of an emergency, accounting for 3/4 and 2/3 of the consultations, respectively (Figure 1). In our study, 70% of the patients who had more than three VOCs at the time of their emergency consultation had SS- or Sβ-type SCD. Eight percent of the children presenting to the ED had a history of stroke (Table 1). Approximately 10% of the children presenting to the ED during the study period had previously undergone a splenectomy. Moreover, 27% of children underwent a cholecystectomy. Half of the children received background HU treatment. A total of 39% of the children received a blood transfusion or exchange transfusion (Table 1). A total of 37% of the emergency consultations were for children with an associated pathology. The most common pathology was asthma (17%), followed by lead poisoning (approximately 8%). VOCs were the most common reason for consultation (67%), followed by infections (12%) (Figure 2). More than half of the VOC-related consultations in this study were associated with severe pain (Visual Analogue Scale (VAS) score, >7). The number of consultations increased from 2012 to 2019, peaked in 2019, decreased from 2019 to 2021, and then increased again in 2022 (Figure 3). There was a decrease in the number of admissions between December and June. The peak in admissions was reached during the long dry season (Figure 4) and in 2019 (Figure 3). Thirty-seven percent of emergency admissions were to pediatric wards. The risk factors for hospitalization following an emergency consultation were an age between 5 and 10 years and a severe form of SCD, characterized by complications (>3 VOCs per year, previous history of ACS, transfusion, stroke, or cholecystectomy). The risk factors for hospitalization are listed in Table 2. All the patients had good outcomes. There were no deaths.

## 4. Discussion

This study aimed to obtain estimates on the characteristics of ED visits by children with SCD in French Guiana. The estimated average annual number of ED visits increased by almost 20% from 220 to 264. This increase in the number of ED visits was consistent with the findings of other studies [8,9,10,11]. This may be partly due to the increased use of observation units, as patients become better educated on how to manage their illness. The average ED waiting time for children with SCD decreased from 2 h to 45 min, with a difference of approximately 1 h and 15 min waiting in the ED. This reduction in the waiting time for children with SCD could be explained by the introduction of a dedicated consultation circuit and improvements in the training of health professionals. In this study, we found statistically significant age-related differences in the ED visits. The median age at admission to the ED was 8 years, and the majority of patients had the SS phenotype. The results of this study also suggested important factors associated with conventional hospitalization after ED admission. The SS phenotype and a history of ACS were factors that led to hospitalization after ED admission. We also found an increase in admissions during the long dry season, from July to November. The role of the climate in the occurrence of VOCs has been previously described in French Guiana and elsewhere [12,13,14].

Consistent with previous studies, pain remained the most commonly reported reason for patient consultations. More than half of SCD-related consultations in this study were associated with severe pain (VAS score, >7). Opioid analgesics are essential in the management of SCD-related pain, with an estimated 40% of patients with SCD taking opioids in a given year. Healthcare providers working in the emergency department, therefore, need to be prepared to assess and manage SCD-related pain, which is particularly important given the documented systemic inequalities in pain management in patients with SCD. Healthcare providers often associate patients with VOCs with opioid abuse and addiction, and previous studies have shown that the rise of the opioid epidemic in the US has led to increased stigma and barriers to opioid access for people with SCD [15,16,17,18,19,20,21]. Given the high level of pain associated with sickle cell emergency visits in this study, the finding of an average wait time of approximately 45 min to see a healthcare professional is particularly high, although efforts continue to reduce this wait time. Nevertheless, evidence-based guidelines recommend that patients with SCD experiencing acute pain should receive prompt analgesic treatment within 30 min of emergency triage or within one hour of emergency registration, followed by frequent reassessment of their pain every 30–60 min. For people with VOCs associated with severe pain, the guidelines recommend rapid initiation of parenteral opioid therapy within the same timeframe, but with pain reassessment every 15–30 min until the pain is controlled. Although we were not able to directly assess the analgesic administration time from the data analyzed in this study, it is likely that the waiting time, in addition to the analgesic administration time, may have led to an increase in the number of patients experiencing pain. It is important to address suboptimal ED waiting times for people with SCD, particularly for children experiencing pain.

Fever was the second most common cause of ED visits among children with SCD. Therefore, urgent medical assessment is recommended for children with SCD and fever. It is a standard protocol for collecting blood cultures and administering antibiotics to all febrile children (>38.5 °C) with SCD. However, there is often a delay in administering these antibiotics in the emergency department. Advances in communication between pediatric hematologists and ED physicians have reduced the time to antibiotic administration in febrile children with SCD [22].

Acute symptomatic anemia was the third most common cause of ED consultation. Patients were treated with a simple transfusion to restore their hemoglobin levels to the baseline. Patients with sickle cells are frequently given transfusions and are at a risk of developing alloimmunization. Therefore, transfusions were performed only under certain conditions. If it is not possible to avoid transfusions, it is recommended that the most compatible units of red blood cells are given [23]. Again, communication between the pediatric hematologist and the emergency physician allows the child to be given a transfusion in safe conditions [22].

The results of our study have several important implications for the clinical care of children with SCD. Children with the most severe phenotypes presented most frequently to the ED. These results are comparable to those published by other authors in both developed and developing countries [24,25,26,27,28]. Knowledge on the causes of ED admissions and risk factors for hospitalization is essential to better guide the prevention of complications and improve the protocol for the selection of disease-modifying therapy.

Another important issue is how emergency departments are organized within hospitals. A number of studies have shown that sickle cell patients often receive poor care in the ED due to a lack of staff familiarity with the disease and suboptimal pain management, including the management of acute splenic sequestration. Patients are sometimes stigmatized. They are told that they are coming to the hospital to obtain a mixture of nitrous oxide and oxygen or morphine, which is the main drug administered in such circumstances, and that they are experiencing withdrawal symptoms [24,25]. Some health professionals say that patients are faking pain because their faces do not reflect the intensity of the pain. In French Guiana, we decided to improve the organization of emergency care for sickle cell patients by setting up a 24 h on-call service, which means that any emergency physician can call on this service if there are difficulties in dealing with a sickle cell patient in the ED. This will enable them to provide better and safer care for children with SCD, as certain diagnostic or therapeutic errors can lead to serious complications, such as ACS or stroke.

Improved SCD management of sickle cell emergencies contributes to the quality of overall care provision. However, the key to improving the overall quality of life for patients with the disease is the indication of a disease-modifying therapy [29,30]. The current trend is to start hydroxyurea (HU) treatment earlier [31]. This treatment is now widely available and should be included in the World Health Organization’s (WHO) list of essential medicines, which would also benefit developing countries, where the prevalence of SCD is the highest [32,33].

The main limitation of this study was its retrospective nature, which did not allow us to describe the time from admission to analgesic administration. However, this study was monocentric. Of course, it only concerns the Cayenne Hospital, which is the reference center. The other two ED in French Guiana were excluded. Nevertheless, the external validity of this study is good because it reflects the reality in our region. In addition, the sickle cell on-call service is regional and, therefore, benefits all hospitals in French Guiana. Local prevention and care centers are also beneficial.

## 5. Conclusions

In conclusion, this study shows that the number of ED visits by children with SCD continues to increase and that children with SCD have a higher risk of being admitted to the hospital after a visit to the ED. The management of pain and fever is often delayed. This leads us to suggest that systematic prior communication between the pediatric hematologist and the ED physician should occur. However, the indications for disease-modifying therapy should be reviewed and, in particular, prescriptions should be extended to HUs to reduce these admissions. We emphasize the need for continued medical education for ED physicians and nurses. However, there is a need to define the best practices for the management of children with SCD presenting to the ED with fever.

## Figures and Tables

**Figure 1 diseases-13-00142-f001:**
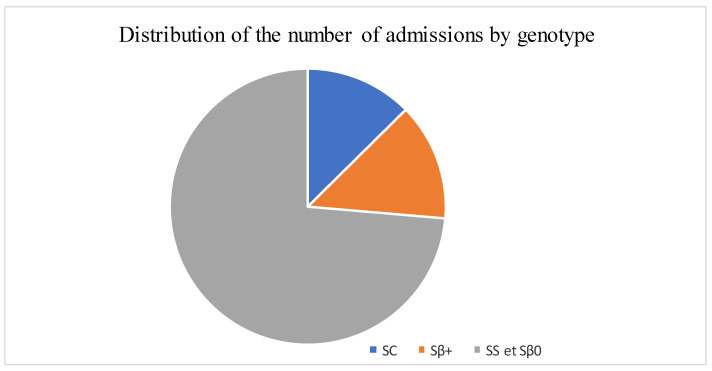
Distribution of the number of admissions by genotype.

**Figure 2 diseases-13-00142-f002:**
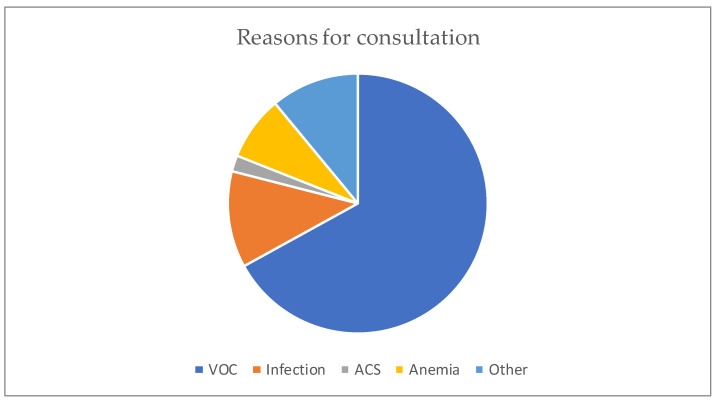
Distribution according to the reason for the consultation.

**Figure 3 diseases-13-00142-f003:**
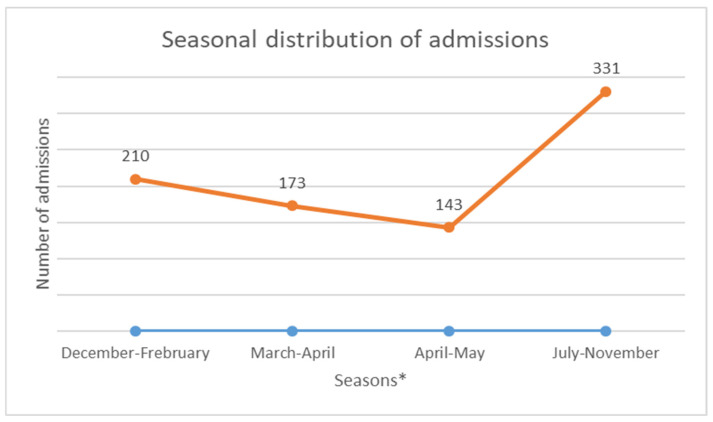
Annual change in the number of consultations. * December to February is the short rainy season. March to mid-April is a short summer season. The main rainy season occurs from mid-April until the end of June. The dry season lasts from mid-July until the end of November.

**Figure 4 diseases-13-00142-f004:**
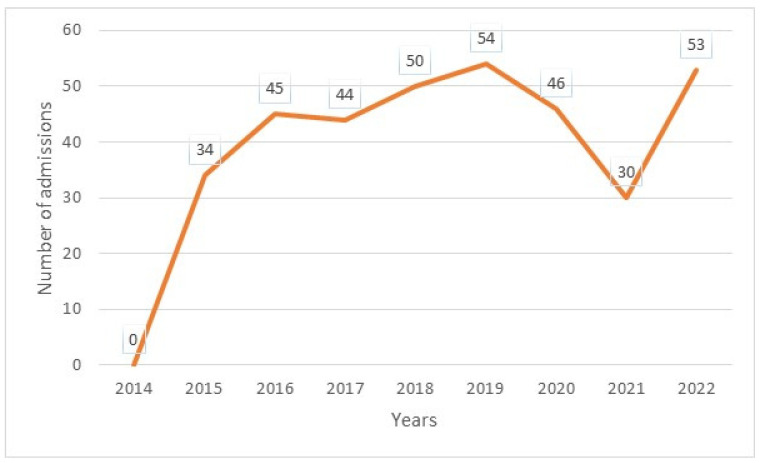
Annual number of emergency admissions.

**Table 1 diseases-13-00142-t001:** Characteristics of sickle cell patients admitted to emergency department.

Variables	Number (%)
Age distribution	
<5 years	109 (13%)
5–10 years	217 (25%)
>10 years	531 (62%)
Gender	
Famale	441 (51)
Male	416 (49)
Sickle cell genotype	
SC	107 (12)
Sß+thalassemia	117 (14)
SS/Sß°thalassemia	626 (74)
History	
>3 VOCs per year	585 (71)
Sepsis	234 (28)
ACS	239 (29)
Transfusion/TE	318 (39)
Stroke	64 (8)
Splenectomy	83 (10)
Cholecystectomy/cholelithiasis	222 (27)
Associated pathology	310 (37)
First quarter	233 (27)
Second quarter	218 (26)
Third quarter	189 (22)
Fourth quarter	217 (25)
Emergency department visit reason	
Pain (VOC)	573 (67)
Fever (infection)	99 (12)
Pallor (anemia)	70 (8)
Difficulty breathing (ACS)	19 (2)
Other *	93 (11)
Hospitalizations	285 (37)
Length of hospital stay	
<5 days	239 (84)
≥5 days	46 (16)
Definitive diagnosis	
Vaso-occlusive crisis	513 (68)
Anemia or splenic sequestration s	98 (13)
Sepsis/infection	104 (14)
Acute chest syndrome	21 (3)
Other	24 (3)
Treatments	
IV Hydration	858 (100)
Morphine	85 (11)
Antibiotics	113 (15)
Oxygen	40 (5)

VOC: vaso-occlusive crisis; ACS: acute chest syndrome. Other *: other raison for Emergency department visit such as skin infection, acute gastroenteritis.

**Table 2 diseases-13-00142-t002:** Multivariate analysis of risk factors for hospitalization in children with sickle cell disease admitted to emergency departments.

Variables.	Children Hospitalized	Children not Hospitalized	AOR	*p*
	Number (%)	Number (%)		
Age distribution			0.7 [0.6–0.9]	0.01
<5 years	44 (42)	60 (58)		
5–10 years	93 (46)	111 (54)		
>10 years	148 (33)	306 (67)		
Gender			0.8 [0.6–1.1]	0.2
Famale	157 (40)	231 (60)		
Male	128 (34)	246 (66)		
History				
>3 Annual VOCs	216 (41)	313 (59)	2 [1.3–3.1]	0.001
ACS	95 (44)	116 (56)	1.6 [1.1–2.5]	0.02
Transfusion/TE	95 (32)	203 (68)	0.7 [0.4–1]	0.07
VCA	9 (14)	55 (86)	0.5 [0.2–0.9]	0.04
Cholecystectomy/cholelithiasis	59 (26)	154 (74)	0.5 [0.4–0.8]	0.03

## Data Availability

Data supporting this research will be available upon reasonable request.

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
