# Peer review of "Emergency Presentations of Pediatric Sickle Cell Disease in French Guiana"

_diseases, 2025, doi:10.3390/diseases13050142_

Round 1
Reviewer 1 Report
Comments and Suggestions for Authors
In the current study, Dr. Djomo and colleagues structured and analyzed clinical data on young patients with sickle cell disease admitted and treated at the Cayenne Hospital over a 9-year period from 2014 to 2022. The authors analyzed a number of key parameters associated with the disease, including sickle cell genotype, initial reasons for referral, influence of season on emergency admission, etc., from which they identified a number of risk factors for hospitalization (e.g., more than 3 annual vasoocclusive crises or the presence of acute chest syndrome). The results described by Djomo et al. are of practical interest for optimizing the management of sickle cell disease not only in French Guiana, but also in other countries where the disease is prevalent. On the basis of their findings, the authors argue for the need to organize a 24-hour hotline to help emergency physicians determine optimal treatment regimens for children with SCD, which could significantly improve the effectiveness of their work.
Given the prevalence and severity of sickle cell disease in children worldwide, this paper may be published in the journal Diseases after some revisions:
Major comments:
Please elaborate on the statistically significant difference between groups shown in Table 2 in more detail. What specific groups were compared between groups? What criterion was used to calculate the p-value?
Dear authors, please give more information on SS and Sbeta genotypes of SCD. What changes underlie them?
Minor comments:
Line 153 - please decipher the abbreviations SS and Sbeta0
Line 154 - please change 75% to 3/4 according to the style used Figures - please (a) remove the titles from the figures as this information is already included in the figure legends, (b) remove the borders around the figures.
Figure 1 - please delete the yellow marker from the legend inside the figure
Figure 4 - as I can see from the image quality, this figure is a photograph taken from a computer monitor. Dear authors, please include the original image in the article.
Table 2 - please change the font to Palatino Linotype as used in the text of the manuscript Please remove unnecessary numbering from the reference list.
Author Response
Responses to the editors and the reviewers
Journal : Diseases (ISSN 2079-9721)
Manuscript ID : diseases-3527669
Dear reviewer
Please accept the revision of our original article, "Emergency presentations of Paediatric Sickle Cell Disease in French Guiana: A tertiary Centre. Descriptive Study" by Narcisse Elenga * , Carine Djomo Fankep , Souam Ngelé Silé e are grateful of the reviewer's comments and have addressed them with the point-by-point response below. We truly believe that their constructive comments have further improved our article.
Could you please reorganize the abstract with the following structure:
Background/Objectives: xx; Methods: xx; Results: xx; and Conclusions: xx?
We have organised the abstract structure
Comments and Suggestions for Authors
In the current study, Dr. Djomo and colleagues structured and analyzed clinical data on young patients with sickle cell disease admitted and treated at the Cayenne Hospital over a 9-year period from 2014 to 2022. The authors analyzed a number of key parameters associated with the disease, including sickle cell genotype, initial reasons for referral, influence of season on emergency admission, etc., from which they identified a number of risk factors for hospitalization (e.g., more than 3 annual vasoocclusive crises or the presence of acute chest syndrome). The results described by Djomo et al. are of practical interest for optimizing the management of sickle cell disease not only in French Guiana, but also in other countries where the disease is prevalent. On the basis of their findings, the authors argue for the need to organize a 24-hour hotline to help emergency physicians determine optimal treatment regimens for children with SCD, which could significantly improve the effectiveness of their work.
Given the prevalence and severity of sickle cell disease in children worldwide, this paper may be published in the journal Diseases after some revisions:
Major comments:
Please elaborate on the statistically significant difference between groups shown in Table 2 in more detail. What specific groups were compared between groups? What criterion was used to calculate the p-value?
Thank you for this comment. We have completed it in the methods section.
Dear authors, please give more information on SS and Sbeta genotypes of SCD. What changes underlie them?
Thank you for this comment. We have completed it in the introduction section.
Minor comments:
Line 153 - please decipher the abbreviations SS and Sbeta0
OK done
Line 154 - please change 75% to 3/4 according to the style used Figures - please (a) remove the titles from the figures as this information is already included in the figure legends, (b) remove the borders around the figures.
OK done
Figure 1 - please delete the yellow marker from the legend inside the figure
OK done
Figure 4 - as I can see from the image quality, this figure is a photograph taken from a computer monitor. Dear authors, please include the original image in the article.
OK done
Table 2 - please change the font to Palatino Linotype as used in the text of the manuscript Please remove unnecessary numbering from the reference list.
OK done
Reviewer 2 Report
Comments and Suggestions for Authors
The authors have presented an interesting work summarizing the advice in the emergency departments of French Guiana for children with SCA. I like the work very much, the authors have neatly described the limitations in the manuscript. I have a few comments - basically, the abbreviation SCD is associated with sudden cardiac death... and in the context of the work, it would probably be better to use the alternative SCA. But I absolutely do not insist here. The manuscript is coherent. I am only wondering about one thing. When the title gives the place, it means that we are either dealing with a special place different from the "neighbors" or it is a separate problem due to some problem. The discussion and information about the organization of medical care shed some light on this matter. However, it is superficial. I would expect this text in the introduction. It would be good to include information about the epidemiological situation of the region in the introduction, then Guyana vs. neighbors. Similarly, I would add a few sentences about the organization of health care, the presence of reference centers (Guyana vs. neighbors). A few sentences about the organization of health care. This issue seems to me to be equally interesting and worth presenting. I think that such an expansion of the introduction will make the manuscript interesting and readable for hematologists, emergency room doctors and pediatricians.
Author Response
Responses to the reviewer2
Journal : Diseases (ISSN 2079-9721)
Manuscript ID : diseases-3527669
Dear reviewer
Please accept the revision of our original article, "Emergency presentations of Paediatric Sickle Cell Disease in French Guiana: A tertiary Centre. Descriptive Study" by Narcisse Elenga * , Carine Djomo Fankep , Souam Ngelé Silé e are grateful of the reviewer's comments and have addressed them with the point-by-point response below. We truly believe that their constructive comments have further improved our article.
The authors have presented an interesting work summarizing the advice in the emergency departments of French Guiana for children with SCA. I like the work very much, the authors have neatly described the limitations in the manuscript. I have a few comments - basically, the abbreviation SCD is associated with sudden cardiac death... and in the context of the work, it would probably be better to use the alternative SCA. But I absolutely do not insist here. The manuscript is coherent. I am only wondering about one thing. When the title gives the place, it means that we are either dealing with a special place different from the "neighbors" or it is a separate problem due to some problem. The discussion and information about the organization of medical care shed some light on this matter. However, it is superficial. I would expect this text in the introduction. It would be good to include information about the epidemiological situation of the region in the introduction, then Guyana vs. neighbors. Similarly, I would add a few sentences about the organization of health care, the presence of reference centers (Guyana vs. neighbors). A few sentences about the organization of health care. This issue seems to me to be equally interesting and worth presenting. I think that such an expansion of the introduction will make the manuscript interesting and readable for hematologists, emergency room doctors and pediatricians.
As far as the SCD is concerned, this is the conventional abbreviation.
In the introduction we have added some information about French Guiana's neighbours.
We also provided more information about the organization of medical care in our center.
We wish that our revised manuscript is now suitable for publication in your journal.
Yours sincerely
Round 2
Reviewer 1 Report
Comments and Suggestions for Authors I am grateful to the authors for their attention to my comments. All comments have been duly considered and corrected. In my opinion, the article is ready for publication after minor corrections to the figures. Figures 1 and 2 - unfortunately, in the revised version of the manuscript, the legends have been moved so that they are not visible. Please correct. Figure 4 - Please provide names for OY and OX axes.Author Response
Responses to the editors and the reviewers
Journal : Diseases (ISSN 2079-9721)
Manuscript ID : diseases-3527669
Dear reviewers,
Please accept the 2nd revision of our original article, "Emergency presentations of Paediatric Sickle Cell Disease in French Guiana: A tertiary Centre. Descriptive Study" by Narcisse Elenga * , Carine Djomo Fankep , Souam Ngelé Silé e are grateful of the reviewer's comments and have addressed them with the point-by-point response below. We truly believe that their constructive comments have further improved our article.
Comments and Suggestions for Authors,
I am grateful to the authors for their attention to my comments. All comments have been duly considered and corrected. In my opinion, the article is ready for publication after minor corrections to the figures.
Thank you so much
Figures 1 and 2 - unfortunately, in the revised version of the manuscript, the legends have been moved so that they are not visible. Please correct. Figure 4 - Please provide names for OY and OX axes.
Thanks for the comments. We have corrected figures 1 and 2 and then figure 4.